# System identifiability in a time-evolving agent-based model

Tal T. Robin[1]*, Jaime Cascante-Vega[1], Jeffrey Shaman[1,2‡]*, Sen Pei[1‡]*

**1** Department of Environmental Health Sciences, Mailman School of Public Health, Columbia University, New York, NY, United States of America, **2** Columbia Climate School, Columbia University, New York, NY, United States of America

‡ JS and SP are joint senior authors on this work.
* tal.robin@columbia.edu (TTR); jls106@cumc.columbia.edu (JS); sp3449@cumc.columbia.edu (SP)

**Data Availability Statement:** Data and code are available at: https://github.com/TalRobin/ABM_Identifiability Data includes the network structure without individual information about patients.

## Abstract

Mathematical models are a valuable tool for studying and predicting the spread of infectious agents. The accuracy of model simulations and predictions invariably depends on the specification of model parameters. Estimation of these parameters is therefore extremely important; however, while some parameters can be derived from observational studies, the values of others are difficult to measure. Instead, models can be coupled with inference algorithms (i.e., data assimilation methods, or statistical filters), which fit model simulations to existing observations and estimate unobserved model state variables and parameters. Ideally, these inference algorithms should find the best fitting solution for a given model and set of observations; however, as those estimated quantities are unobserved, it is typically uncertain whether the correct parameters have been identified. Further, it is unclear what 'correct' really means for abstract parameters defined based on specific model forms. In this work, we explored the problem of non-identifiability in a stochastic system which, when overlooked, can significantly impede model prediction. We used a network, agent-based model to simulate the transmission of Methicillin-resistant staphylococcus aureus (MRSA) within hospital settings and attempted to infer key model parameters using the Ensemble Adjustment Kalman Filter, an efficient Bayesian inference algorithm. We show that even though the inference method converged and that simulations using the estimated parameters produced an agreement with observations, the true parameters are not fully identifiable. While the model-inference system can exclude a substantial area of parameter space that is unlikely to contain the true parameters, the estimated parameter range still included multiple parameter combinations that can fit observations equally well. We show that analyzing synthetic trajectories can support or contradict claims of identifiability. While we perform this on a specific model system, this approach can be generalized for a variety of stochastic representations of partially observable systems. We also suggest data manipulations intended to improve identifiability that might be applicable in many systems of interest.

**Funding:** JS received funding in the form of a grant from the Foundation for the National Institutes of Health (R01AI163023) and the Andrew and Corey Morris-Singer Foundation. SP received funding in the form of a grant from the Council of State and Territorial Epidemiologists (NU38OT00297). JS and SP received funding in the form of a grant from the National Center for Emerging and Zoonotic Infectious Diseases (U01CK000592). The funders had no role in study design, data collection and analysis, decision to publish, or preparation of the manuscript.

## 1. Introduction

Evaluation and optimization of novel intervention approaches to control infectious diseases in real world settings can be logistically challenging, expensive, and often ethically problematic. Mechanistic models of disease transmission provide an important tool supporting the fight against the spread of infectious diseases, as these systems offer an alternative *in silica* environment for exploring intervention options. It is thus unsurprising that mathematical models are in widespread use in epidemiological research. These models range from compartmental models using aggregated population states to describe transmission dynamics to high dimensional agent-based models [1–14] that attempt to represent the individual mixing structure of a population as it might influence the progression of an epidemic. However, models present their own challenges, as the data needed to constrain model simulations are often sparse and incomplete. For instance, observed numbers of individuals with a particular bacterium might correspond to only a small fraction of the true number colonized. In this circumstance, models must be able to reliably represent the hidden progression of an outbreak based on sparse observations.

A model can adequately predict the progression of an outbreak, i.e., disease transmission dynamics, even if mis-specified in some of its assumptions, provided it captures the relationship between observations and state variables of interest. However, the accuracy of those predictions is typically highly dependent on good estimation of model system parameters. Using parameters across different models (for example, using the transmission rate estimated in a previous modeling study) can be problematic because there may be significant differences of same parameters in different systems or under different model assumptions.

To find the parameters relevant for a given model, many researchers augment their models with data that are used to inform parameter selection [15–20]. The data are often direct observations of the system state (diagnostic data) or data describing conditions influencing the system state (e.g., meteorological conditions or patient location data). Using these data, infectious disease model parameters that give the best agreement between the observed data and model predictions can be inferred with methods such as Bayesian filters [21–23] or gradient descent methods [24, 25]. Underlying this parameter estimation is the issue of parameter identifiability, which, if not addressed properly can distort prediction and system certainty [26, 27]. In essence, identifiability poses the following question: assuming a model is correctly specified, and given the observations in hand, can the true parameters be reliably estimated? While the "true" parameters are typically not observed and remain unknown, system identifiability can be explored by creating synthetic observations using a given model and set of parameters, and then testing whether an inference method, equipped only with the synthetic observations, can consistently estimate the correct parameters.

The issue of identifiability has been explored in a number of fields [28–30] including infectious disease modeling. For instance, Sauer et al. demonstrated that parameters of SEIR (susceptible-exposed-infected-recovered) epidemic models are intrinsically unidentifiable early in an epidemic and derived an unidentifiability manifold in the parameter space that consists of parameters indistinguishable from early epidemic curves [27]. Gallo et al. showed that lack of practical identifiability may hamper reliable predictions in compartmental COVID-19 epidemic models and developed a method to characterize different regimes of identifiability [26]. However, existing studies have predominantly focused on compartmental epidemic models described by ordinary differential equations. Identifiability for highly stochastic and high-dimensional agent-based models, which are increasingly used in real-world applications [31–37], has not been systematically evaluated.

Indeed, for deterministic models, the question of identifiability is more straightforward and can be defined as either structural non-identifiability (where more than one set of parameters

result in the exact same outcome) or practical non-identifiability (where lack of sufficient data or excess noise disrupt identification of most likely parameters). For such deterministic systems, there is a well-established literature on how to identify and, in certain circumstances, even fix issues of identifiability [38, 39]. However, many disease systems are stochastic in nature and better suited for stochastic simulation, often using agent-based models. A stochastic model will produce a distribution of possible trajectories with the same parameters and initial conditions; conversely, the same trajectory might result from a range of different sets of parameters. Such overlap of trajectories produced by different parameters imposes a further challenge when attempting to identify system parameters.

Here we explore the issue of identifiability using a stochastic, agent-based transmission model of Methicillin resistant Staphylococcus Aureus (MRSA), an antimicrobial resistant pathogen prevalent in community and healthcare settings, across 4 hospitals in New York City. Using location data of patients within hospital, we attempted to find the model parameters that best reproduce the number of observed positive MRSA cultures. Specifically, we used an Iterated Ensemble Adjustment Kalman Filter (EAKF) to infer two free parameters: the patient-to-patient transmission rate, β, and the probability a new patient is colonized, γ. We show that while using the EAKF on a specific observed trajectory gives a reliable answer (in the sense that it does not change between runs and appears to converge), using it on different synthetic trajectories derived from the same parameter combination yields different inferred parameters. In addition, we show that the likelihood of observing a particular trajectory given a set of parameters,

*Pr(trajectory|β,γ)*, does not have a global maximum, but instead has a ridge of equally likely values, representing an interplay between the different colonization processes (importation vs. nosocomial transmission) that yields indistinguishable trajectories. We demonstrate the effect of this non-identifiability on the ability to predict hidden variables and assessment of the efficacy of interventions.

## 2. Methods

### 2.1 Data

Admission and discharge records for 278,522 distinct patients from 4 hospitals in New York City, New York, USA were used in this study. These data span 1399 consecutive days (02/01/2009 to 11/30/2012). For each patient, records include patient ID, location, admission, and discharge time. The records also include culture data from specimens obtained during hospitalizations, and whether MRSA, among other pathogens, was identified. Across the 4 hospitals, patients were distributed among 315 wards of varying size; ward size, a measure used in the dynamic model, was defined as the mean daily occupancy of the ward (ward size distribution followed a power law with mean 8.7 patients). Patients spent on average 6.3 days in hospital per visit. In total, 1845 positive MRSA cultures were recorded (excluding repeat positive tests obtained during the same hospitalization).

### 2.2 Model

We developed a network agent-based model in which each agent represents a patient admitted to the hospital, and interactions exist between patients in the same ward on the same day (Fig 1). The patient state can be either colonized or susceptible and transitions occur between the states in daily increments. Transition from colonized to susceptible (decolonization) occurs at a constant rate $\alpha$[1/day]; however, transition from susceptible to colonized (colonization) occurs for susceptible patients co-located with a colonized patient at the density dependent rate $\beta/N_w$, where $N_w$ is the ward size of ward $w$. We note as $w_i^d$ the ward where patient $i$ was

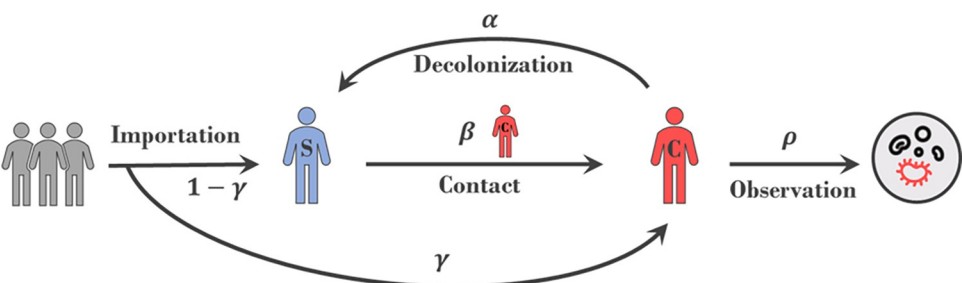

**Fig 1. Schematic representation of the mechanistic model: A new patient introduced to hospital for the first time is colonized with probability $\gamma$.** A susceptible patient in contact with a colonized patient (i.e., in the same ward) has a probability $\beta/N_{w_i^d}$ of becoming colonized each day of contact, where $N_{w_i^d}$ is the number of patients in the ward where patient $i$ stayed on day $d$. A colonized patient can decolonize and become susceptible each day (whether in hospital or not) at probability $\alpha$. A colonized patient is observed (tests positive for MRSA) each day with probability $\rho$.

hospitalized on day $d$, and $\beta$[1/day] is the transmission rate. Each patient is already colonized upon admission with probability $\gamma$ (importation rate). Model dynamics can be summarized by the equation indicating the probability that patient $i$ is colonized on day $d$, $C_i^{d*} \in [0,1]$:

$$C_i^d* = (1-\alpha)C_i^{d-1} + \frac{\beta}{N_{w_i^d}}\left(1-C_i^{d-1}\right)\sum_{j \in w_i^d} C_j^{d-1} + \gamma h_i^d \qquad 0.1$$

Where $C_i^{d-1}$ is equal to 1 if patient $i$ was colonized on the previous day, and 0 otherwise. $h_i^d$ is equal to 1 if patient $i$ was admitted to the hospital for the first time on day $d$ and 0 otherwise.

If a patient is present in more than one location in the hospital network on a given day, he/she is assumed to have contributed to the transmission for each location in inverse proportion to the number of locations visited. That is, if we denote the number of locations patient $i$ visited as $k_i^d$, Eq 1 is rewritten as:

$$C_i^d* = (1-\alpha)C_i^{d-1} + \gamma h_i^d + \sum_m^{k_i^d} \frac{1}{k_i^d}\frac{\beta}{N_{w_{i,m}^d}} S_i^{d-1}\sum_{j \in w_{i,m}^d}\frac{1}{k_j^d}C_j^{d-1} \qquad 0.2$$

where $w_d^{i,m}$ is the $m^{th}$ ward patient $i$ visited on day d.

The colonization status of patient $i$ on day d, $C_i^d$, is determined to be either 1 or 0 with probabilities $C_i^{d*}$ and $(1- C_i^{d*})$ respectively.

Once the colonization status of each patient is determined, each colonized patient is observed (i.e. by positive test result) with probability $\rho$ (See algorithm 1 in the S1 File for more detail).

## 2.3 Inference method

Our intention is to infer a set of parameters that, given our model, would be most likely to produce the time series of observed positive (colonized) patients. For that, we implemented an Iterated Ensemble Adjustment Kalman Filter (EAKF). In this method, we used the sample covariance between an ensemble of parameters and a corresponding ensemble of variables to adjust the distribution of parameters.

To perform this inference, we used an ensemble of model simulations. Each simulation was initiated with a set of parameters $\hat{\Theta}$, as well as a corresponding ensemble of colonization states, $\hat{C}$ and observation state $\hat{O}$, (for each parameter set–a corresponding colonization and observation state for all patients).

Using the model, we progressed the colonization state, $\hat{C}$, daily for each ensemble member (as described in algorithm 1 in the S1 File) over an assimilation period consisting of $d$ days (Algorithm 2 in S1 File). During this integration, we accumulated the number of observed colonized patients, $\hat{O}$, for each ensemble member.

After each assimilation period, we used the Ensemble Adjusted Kalman Filter (Algorithm 3 in S1 File) to update the parameters, $\hat{\Theta}$. Using the updated parameters, we repeated the process over the next time step until the end of our data. At each time step we recorded the values of all parameters sets in the ensemble and took the average over all time step for each ensemble member as the posterior distribution of the inferred parameters.

## 2.4 Likelihood estimation

To estimate the likelihood $\Pr(\theta_i|\bar{O}_j)$, that a given set of parameters, $\theta_i$, produced a given trajectory $\bar{O}_j$, we estimated the probability $Pr(\bar{O}_j|\theta_i)$ that a simulation with parameter set $\theta_i$ will produce trajectory $O_j$

$$\Pr(\theta_i|\bar{O}_j) = \frac{\Pr(\bar{O}_j|\theta_i)\Pr(\theta_i)}{\Pr(\bar{O}_j)} \qquad 0.3$$

We assumed all parameters in range are equally likely. That is, $\Pr(\theta_i)$ is constant. For a given trajectory, therefore:

$$\Pr(\theta_i|\bar{O}_j) \propto \Pr(\bar{O}_j|\theta_i) \qquad 0.4$$

To estimate $\Pr(\bar{O}_j|\theta_i)$ we can simply produce many trajectories using $\theta_i$ as parameters and find the ratio of trajectories that are identical to $\bar{O}_j$. However, due to the stochastic nature of our system the probability of getting exactly the same number of observations in all 99 assimilation periods is extremely small and would require an unreasonable number of trajectories to produce a robust estimation of the likelihood. Therefore, we assumed that the number of observations between different assimilation periods is uncorrelated and calculate the probability of a trajectory as the product of the probability of each assimilation period separately.

$$\Pr(\bar{O}_j|\theta_i) = \prod_{t=1}^{T}\Pr(\bar{O}^t = \bar{O}_j^t|\theta_i) \qquad 0.5$$

Even when estimating each assimilation period separately, a significant estimate of $\Pr(\bar{O}^t|\theta_i)$ is computationally expensive and would give inflated weight to outliers. Assuming the value of $\bar{O}^t|\theta_i$ is normally distributed (an assumption supported by simulations) we approximated the distribution of $\bar{O}^t|\theta_i$ as a discretized normal distribution with the mean, $\mu_{\theta_i}^t$ and standard deviation, $\sigma_{\theta_i}^t$, from the ensemble of synthetic trajectories that were produced using $\theta_i$.

## 2.5 Intervention simulation

In a non-identifiable model, the observed state can be adequately explained by different model parameters; however, this non-identifiability is only of practical importance if the different possible parameters predict a significant difference in quantities of interest. In our model, we show that using different parameters yields different hidden variables, like the total number of colonizations (Fig 4). A more salient issue is to explore the effect of a known change imposed on the system with multiple sets of possible parameters and whether those differences are

associated with significant changes in observable states, even if the unperturbed system is indistinguishable. This has special relevance in public health where evaluating the effects of interventions is often the goal of a model. a goal.

To explore this issue, we imposed an intervention in the model that causes a proportional decrease in the rate of transmission between patients (i.e. β is reduced such that after the intervention β = kβ). This can be seen as a simplistic representation of enacting hospital wide control policies, such as cleaning, which reduce the probability that colonized patients encounter and infect a susceptible patient.

We can then calculate the effect of the intervention by measuring the difference in observations with and without the intervention.

We do this by comparing pair of trajectories that are identical up to a certain time point (we used 700 days) after which we progress one of these trajectories with the altered transmission rate (kβ) and the other with the unaltered rate (β). We then calculate the difference in total observed cases between the two trajectories, by which we ensure the difference is only due to the intervention. We repeat this exercise running 300 pairs and calculate the mean and variance of the differences. These results are used to estimate the cost-effectiveness of an intervention. By calculating this for different possible parameter sets we examine whether parameter differences produce a difference in the measurable effect of the intervention.

## 3. Results

Our model system leverages hospital records from the New York Presbyterian hospital (NYPH) system (n = 278,522 patients, 4 hospitals). For each patient the data indicates daily location, i.e., ward, within the hospital. In addition, a small fraction of the patients (1845, about 0.7%) have culture records that were positive for MRSA. This number is below the expected number of patients colonized with MRSA (about 5% in the hospital population based on prior studies [40]). To understand the unobserved progression of MRSA colonization in the hospital we constructed a mechanistic model that describes the importation of MRSA to the hospital by colonized individuals and nosocomial transmission between patients in the same ward during the same day (see Fig 1 in methods). Dynamic progression of the model is dependent on four free parameters; β –the nosocomial transmission rate γ –the importation probability α –the decolonization rate and ρ –the daily probability a colonized patient is detected. Using these parameters, we can simulate the progression of colonization among patients and the corresponding number of positive cultures we would expect to observe, which can be compared to actual observations. Specifically, trajectories are defined as the aggregation of the number of observed positive MRSA cultures in each two-week period over the span of 1399 days for which we have data. Due to model stochasticity, each simulation of the model produces a different trajectory, even if run with the same parameters and initial conditions. Therefore, we look for the set of parameters for which the *probability* of simulating the observed trajectory is highest.

We calibrated the agent-based model to the numbers of MRSA positive cultures observed every two weeks in NYPH. To simplify the analysis, we will assume knowledge of two parameters, α and ρ using published empirical studies and only try to infer β and γ. For each simulation, we started by drawing values of β and γ from independent uniform distributions over a wide range of possible values (0–0.1 [1/day]]) and drew the colonization status of each patient already admitted on the first day of simulation with probability to be colonized γ. We progressed the colonization status of the patients in the system each day for a period of 14 days and summed the number of positive cultures for that period. An ensemble of 300 simulations was integrated. We used the EAKF (see methods) to update the values of β and γ for each

ensemble member based on the covariability of these parameters with the observed state variable (simulated number of positive MRSA cultures). We repeated this process over successive 14-day periods to obtain, theoretically, a distribution of parameters with an improved fit to observations. Once the simulation and updating was completed, we averaged the values of parameters β and γ for each ensemble member across the entire time series. This posterior ensemble was then used as the prior for another iterative assimilation of the same time series [41].

Fig 2 shows how the posterior distributions for γ and β change over 20 iterations. Parameter values stabilize around a solution of β = 0.015 and γ = 0.043.

The quality of these estimates was further tested by using the inferred parameters in simulation. Due to the stochastic nature of the model system, the simulated trajectories vary even when using the same parameters. The distribution of trajectories generated from the inferred parameters (Fig 2C) is centered around the observed trajectory which is mostly within the 90% confidence interval of simulations. By its nature, the EAKF inference will gravitate towards

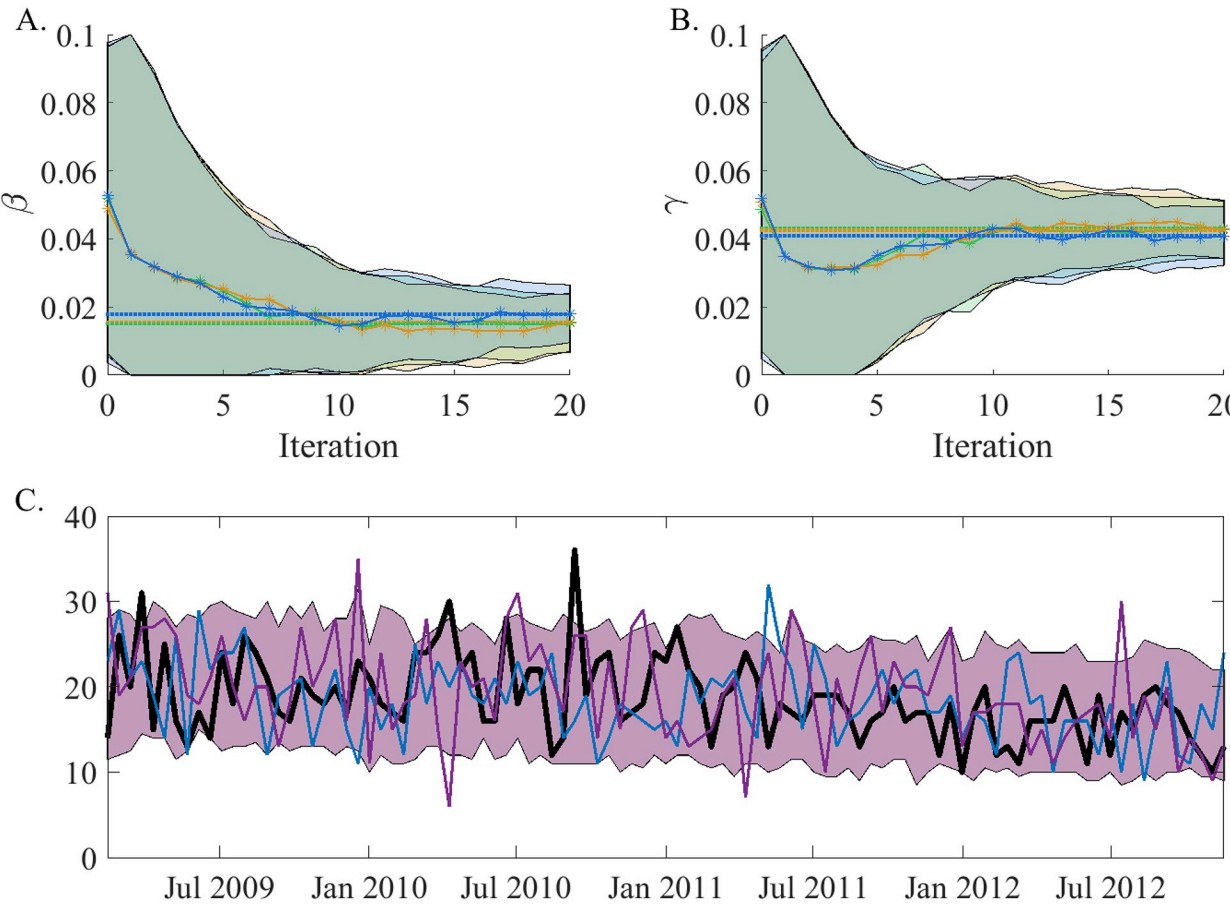

**Fig 2.** Inference results: Panels A and B) Solid lines with stars show the mean value of the posterior distribution of *β* and *γ* respectively, at the end of each itteration of the EAKF. Each color (red, orange and green) represents a different realization of the inference process. The shaded area represents the ensemble 90% confidence interval around the mean of the parameter values across all time steps. The inference was made using the hospital record observations. Dashed lines indicate the values at the final iteration. Values stabilize around *β* = 0.015 and *γ* = 0.043 (where the values for *α* = 1.5/365 and *ρ* = 0.016 were assumed). **Panel C)** Synthetic trajectories simulated using the inferred parameters. The black line is the observed trajectory taken from hospital records and used to perform the inference; the red and blue lines are two synthetic trajectories chosen at random from 300 simulations. The shaded area shows the 90% confidence interval around the mean of the number of observations at each time step for all 300 synthetic trajectories.

parameter values which are more likely to produce agreement with the observations; however, this process might fail to identify the unique result with the highest likelihood. For a high-dimensional, agent-based system, the likelihood landscape may have a complex structure with multiple local maxima. The stochastic noise associated with the dynamics can also obscure the landscape. It is therefore crucial to develop a method to discern whether the optimal solution has been identified.

Calculating the likelihood of all possible parameter combinations at sufficient density is not feasible for many models; however, for our model there are only two free parameters and scanning over this parameter space is possible. Therefore, we performed a likelihood analysis over a large surface of parameters (Fig 3A).

Based on the likelihood landscape, the majority of parameter combinations in the parameter space can be excluded due to low likelihood. However, the landscape is characterized by a narrow ridge where likelihood is similarly high. This ridge can be approximated by a line equation $\beta = 0.0623–1.079\gamma$. The likelihood along the ridge is almost constant for $\gamma > 0.03$ (Fig 3B), indicating a lack of distinction among parameter combinations. The fact that these parameters fall on a line implies an interplay between two processes: a reduction of importation rate combined with an appropriate increase of nosocomial transmission rate produces simulated trajectories that are similarly and indistinguishably well-fit to observations. This finding indicates that the system parameters are not fully identifiable.

While different points along the ridge represent scenarios where the expected observations are indistinguishable, the underlying realities they represent might be significantly different. Fig 4 shows three important metrics for four different parameter combinations along the ridge. As expected, the number of observations is very similar, however, the composition of the source of colonization varies from all colonizations imported to almost 30% of colonizations resulting from nosocomial transmission (Fig 4C). For a public health expert attempting to develop targeted interventions, these two scenarios, which equally account for the

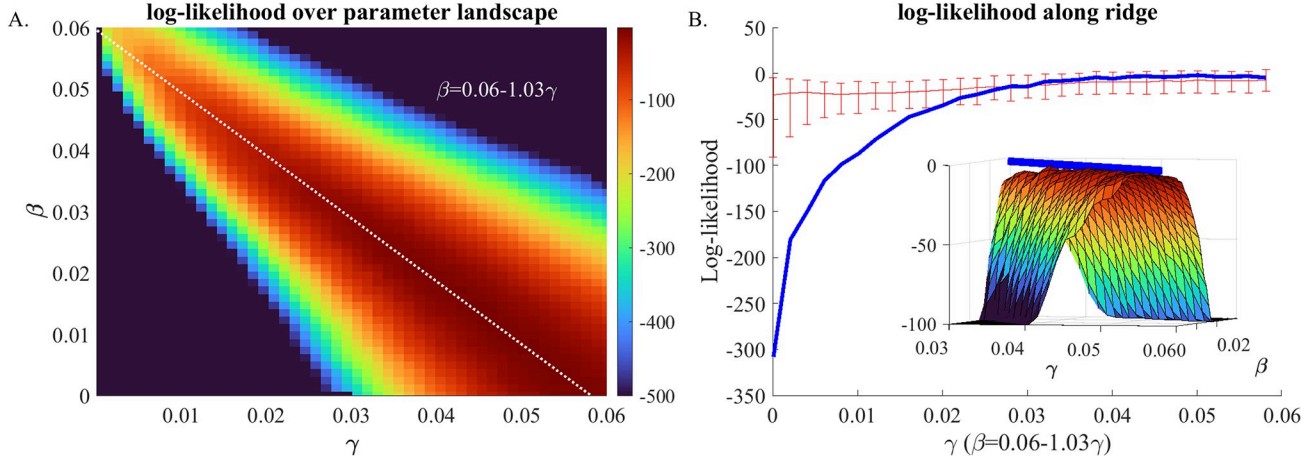

**Fig 3.** Likelihood landscape: A) Heat map of the Log-likelihood of the observed trajectory, $Pr(T_{observed}|\gamma,\beta)$, across the parameter landscape (see S1 File) derived from an ensemble of 300 simulations. Each simulation is a realization of the model using the values of $\gamma$ and $\beta$ indicated on the x and y axis respectively. Dashed white line represents a weighted linear fit of $\beta$ as a function of $\gamma$, where the weight of each coordinate is given by its likelihood. The resulting line equation is indicated above the line in white. Panel B) Likelihood along the ridge. Blue line shows likelihood score of observing the real trajectory given parameters along the ridge. For $\gamma > 0.025$ the likelihood flattens. The likelihood landscape is characterized by a "ridge" along which the slope is very flat compared to the area perpendicular to the ridge. Red lines show the distribution (middle line is the mean; error bars show 90% confidence interval) of the likelihood, $P(T_{i,\gamma,\beta}|\gamma,\beta)$, calculated for each of 200 synthetic trajectories $T_{i,\gamma,\beta}$ generated with the same parameters (i.e., the expected likelihood for stochastic trajectories generated using these parameters). For small values of $\gamma$, the blue line falls below the CI, indicating that the observed trajectory is unlikely compared to the trajectories simulated with those parameters.

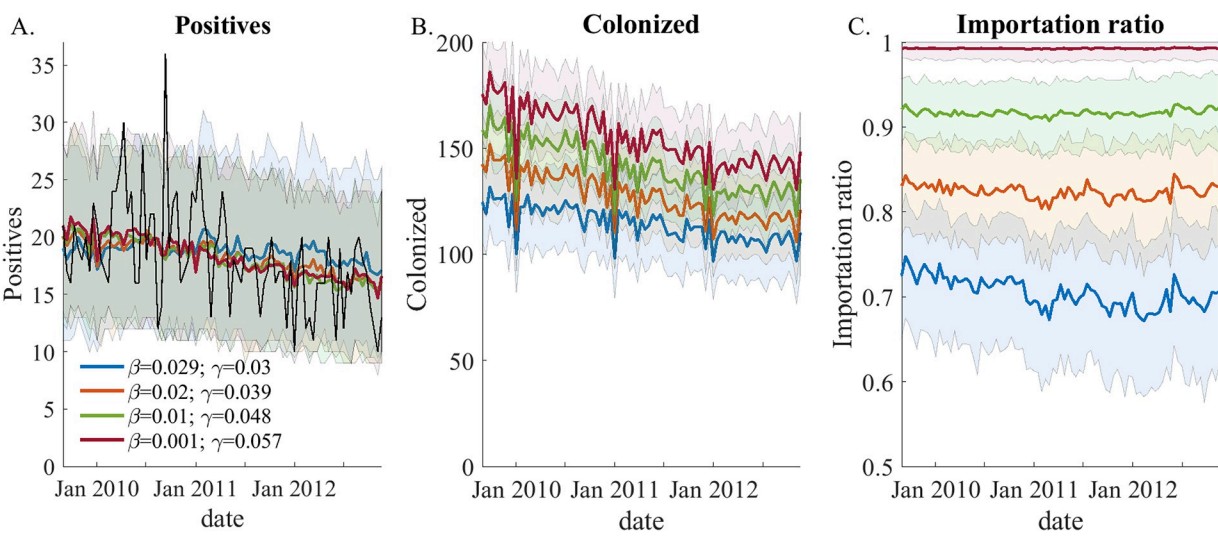

**Fig 4. Different realities: Simulations using 4 parameter combinations that reside on the ridge of equal likelihood described in Fig 3: Blue —β = 0.029, γ = 0.03; orange—β = 0.02, γ = 0.04; green—β = 0.01, γ = 0.05; red—β = $10^{-3}$, γ = 0.06; For each parameter combination simulation was repeated 300 times.** Colored solid lines and shaded areas represent the mean and 90% CI of the variable trajectories. **Panel A)** shows the number of observed positive cultures. The black line shows the number of positive results for the hospital data. All 4 parameter combinations produce an indistinguishable prediction of this variable. **Panel B)** shows the average number of total colonizations. Even though the daily observation ratio is the same (ρ = 0.016) and the number of observations is the similar, the underlying number of colonized patients decreases with increasing β. This illustrates that when more colonizations are due to nosocomial transmission they are more likely to be observed. **Panal C)** shows the average ratio of new cases between importation and nosocomial infections for the ensemble shown in panel A. This ratio also decreases with increasing β.

observations, could point to different conclusions. Even the total number of underlying colonizations, which might be expected to be proportional to the number of observations, changes by 80% along the ridge (Fig 4B). This variation manifests because patients who are hospitalized for a short time are just as likely to be colonized upon admission but are less likely to become infected and be observed during a shorter hospital stay.

The misrepresentation of the true system state is not academic. As can be seen in Fig 5, simulations using different but equally likely parameter combinations can result in very different predictions of intervention effectiveness. In this example, we tested the effect of an intervention that would reduce the nosocomial transmission rate. Such intervention is likely to involve significant investment of resources and public health decision makers might choose to use models to estimate the impact of this intervention on the number of cases. It can be seen that using different parameters which are equally likely to explain observations yield very different predictions of the efficacy of the intervention. This example is, of course, an arbitrary choice of intervention, and is only meant to show that even if observations are not affected by the choice of parameters, these parameters might cause the system to deviate substantially once the system is perturbed.

In more complicated systems with more than two parameters, such parameter space screening is impossible. However, we can learn more about the identifiability of our model-inference system by applying it to a relatively small number of synthetic trajectories. First, we applied the inference method to trajectories simulated with the parameters inferred from the actual culture records. If inference with these synthetic trajectories consistently were to yield the correct parameters, it would strengthen confidence that the system is identifiable because it would show, provided the estimated parameters represent reality accurately, the system can reliably retrieve them using our inference algorithm. However, for our system, as shown in Fig 6,

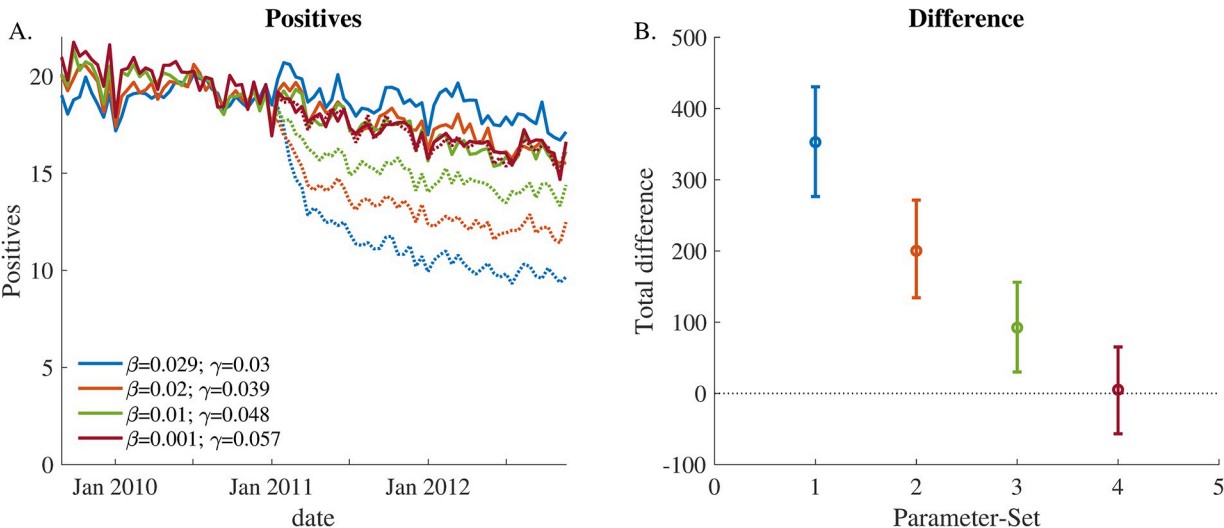

**Fig 5. Different effects of intervention: We simulate an intervention that decreases the nosocomial transmission rate by a factor of 70% ($\beta \to 0.3\beta$) starting from January 2011.** Parameter sets are on the ridge of equal likelihood and are the same as used in Fig 4. **Panel A)** shows the mean trajectories of positive tests with (dashed lines) and without (solid lines) intervention. The trajectories without intervention are indistinguishable, but the effect of the intervention varies significantly. **Panel B)** shows the difference between the total number of positive cases with and without intervention for each of the parameter sets. The difference was calculated from 300 simulations with each parameter combination; circles and error bars show the mean and 90% CI.

inference using 3 different trajectories generated with the same parameters results in different parameter estimates. Despite this variance, all results still fall along the ridge shown in Fig 3.

In a perfect world, additional and more targeted testing can be performed to help differentiate different model parameters; however, many modeled systems are supported by limited data. An informed segmentation of such limited observations can help improve system identifiability by delineating where differences arise between equally likely parameter sets. In our system, it is reasonable to assume that if colonizations are primarily caused by nosocomial transmission (larger $\beta$, smaller $\gamma$), fewer positive results would appear during the first days in hospital, before a patient has contacted potentially infectious peers. Therefore, it may be possible to better distinguish parameters by fitting two time series of positive results: for the first few days following admission and later during the patient stay. In Fig 7 we see how this segmentation enables inference of a substantial reduction of the length of the ridge of equally likely parameters. It must be stated that the results shown in Fig 7 is one example. The success of this method changes dramatically for different parameters and even for different synthetic trajectories of the same parameter set. Further examination, calibration and improvement of this approach are needed and are the subject of ongoing work.

## 4. Discussion

In this work we have explored the identifiability for model parameters in a stochastic agent-based model. We have shown that the model-inference system produced consistent parameter convergence and good agreement with observations by simulations using those parameters. However, the inference system cannot fully identify the unique parameters used for generating synthetic outbreaks. Instead, the estimated parameters tend to lie on a ridge of parameter space with near equal likelihood. Our analysis indicates that the inference system can only identify a region in the parameter space that may include the true parameters with some level of uncertainty.

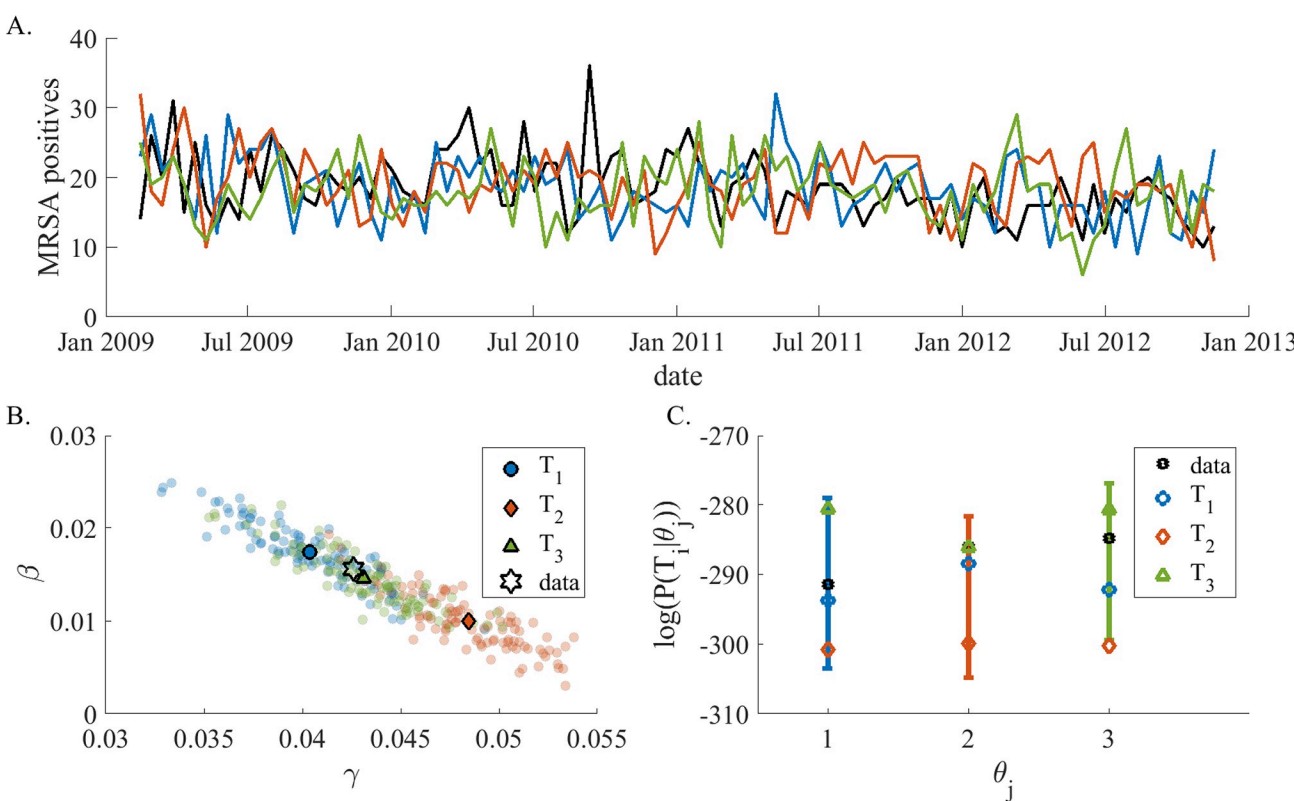

**Fig 6.** Inference over different trajectories from same parameters: **Panel A)** Using the parameters inferred with the real observations ($\beta$ = 0.015 and $\gamma$ = 0.043) we simulated 3 trajectories, $T_1$(blue), $T_2$(orange) and $T_3$(green). The black line is the observed trajectory. **Panel B)** the posterior distribution of each trajectory in panel A produces 3 different solutions: $\theta_1$ = [0.040, 0.017] (blue circle), $\theta_2$ = [0.048, 0.015] (red diamond) and, $\theta_3$ = [0.043, 0.099] (green triangle) corresponding to $T_1$, $T_2$ and $T_3$ respectively. Black star indicates the parameter values used to simulate trajectories $T_1$, $T_2$ and $T_3$. **Panel C)** Likelihood analysis. Each point represents the Likelihood $P(T_i|\theta_j)$ where a blue circle signifies $i$ = 1, orange diamond—$i$ = 2 and green triangle represents $i$ = 3. Lines represents the distribution of $P(T_{\theta_j}|\theta_j)$ where $T_{\theta_j}$ is one of 300 synthetic trajectories that were simulated using the parameter set $\theta_j$. Black stars indicate the likelihood of the observed trajectory $Pr(T_{observed}|\theta_j)$. Each trajectory is almost equally likely to be simulated from any of the inferred parameters.

The identifiability of a system can be affected by four issues. The first is the identifiability of the model, which can be influenced by the complexity of the system (i.e., the number of parameters and interplay between them). Our findings can be reasonably interpreted as an example of this issue. The linear dependance between the inferred parameters along the ridge indicates that an interplay between the parameters may be the cause of unidentifiability. We used a simple model with only two free parameters to illustrate this, but this finding is likely not unique to simple systems and may be exacerbated with additional degrees of freedom. A second aspect affecting identifiability is the degree of stochasticity in the system. In stochastic systems an ambiguity between parameters and observations is inherent. If the spread of observations for simulations generated with different parameters overlap considerably, identifiability might become impossible.

The third component affecting identifiability is the richness of data. The quantity and structure of observations are key for distinguishing, within a given model, between different parameters. If data are sparse or possess too little signal, they may be explained by very different realizations equivalently well. In our example, increasing the quantity of observations improves the probability of finding the correct parameters; however, the system still may not find the unique model parameters due to interplay between parameters and stochasticity (see

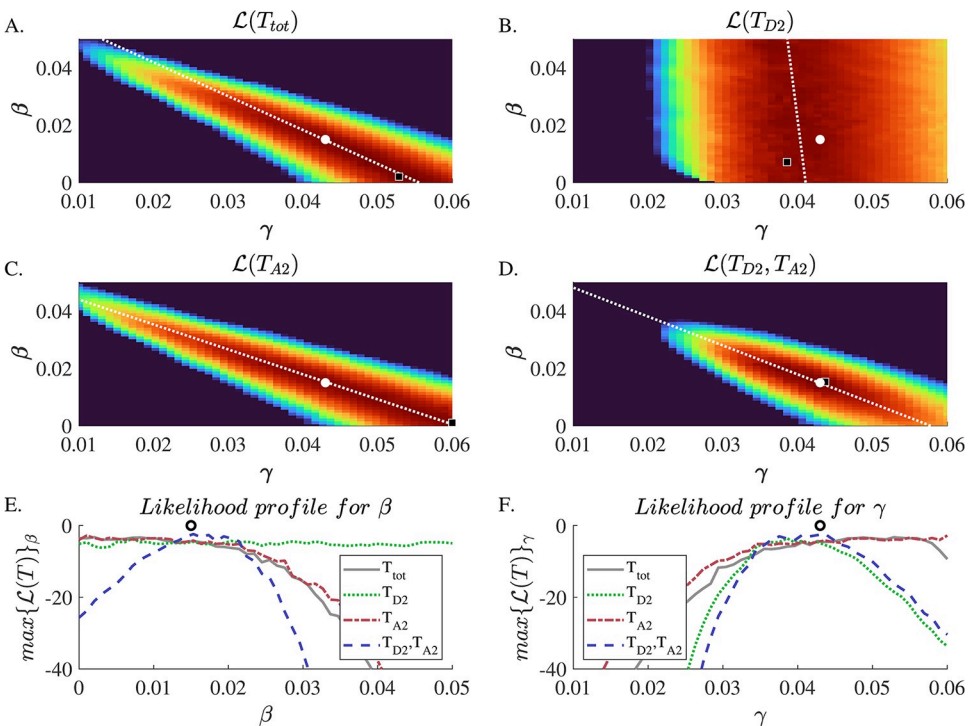

**Fig 7. Panels A-D)** Heat map of the Log-likelihood of the observed trajectory, $\mathcal{L}(T) = Pr(T|\gamma,\beta)$, across the parameter landscape (see S1 File) derived from an ensemble of 200 simulations, similar to Fig 3A. Unlike Fig 3, the likelihood was calculated for a synthetic trajectory with known parameters $\gamma = 0.043$ and $\beta = 0.015$ (marked as the white circle in each panel) A black square represents the maximum likelihood estimate. **Panel A)** shows the likelihood to observe the trajectory of total number of positive tests $T_{tot}$, as in Fig 3. The likelihood landscape is characterized by a long and narrow ridge of similar likelihood. **Panel B)** shows the likelihood of observation of the positive tests that were performed only within the first two days of hospitalization, $T_{D2}$. This likelihood surface is almost flat across different values of beta while showing a relatively sharp peak in $\gamma$. This is to be expected as observations in the first two days are almost completely attributable to imported cases. **Panel C)** shows likelihood of observations of positive tests that were preformed later than the second day of admission, $T_{A2}$. The likelihood landscape appears similar to that of total number of observations. **Panel D)** shows that joint probability to observe both trajectories $T_{D2}$ and $T_{A2}$ simultaneously. It can be seen that the likelihood landscape exhibits a much sharper slope around the peak which is at the correct parameter coordinates. For better visualization of the peaks and plateaus mentioned above, **Panel E and F)** shows the likelihood profiles for $\beta$ ($max\{\mathcal{L}(T)\}_\beta$) and $\gamma$ ($max\{\mathcal{L}(T)\}_\beta$), respectively, where $T$ is $T_{tot}$ (gray solid line), $T_{D2}$ (dotted green line), $T_{A2}$ (dash-dotted red line) or the combination of $T_{D2}$ and $T_{A2}$ (dashed blue line).

S1 File, where the likelihood ridge remains even when the probability of observation, $\rho$, is increased tenfold). The fourth issue is the choice of inference algorithm. There exist many inference algorithms, some of which might be more or less suitable for different systems, and within each algorithm there are often different choices that can be made to improve the performance of the inference (e.g., informed selection of priors, controlled convergence, etc.). However, in systems like the one presented here, a different algorithm would not have produced better parameter inference because there is no theoretical way to distinguish between different parameters given the model, system stochasticity, and data available.

While the quantity of data available for modeling is usually fixed, identifiability might be improved by manipulating the structure of the data to create differentiating observables. As we show in Fig 7, an improvement in identifiability can be achieved by segmenting observation according to length of stay. Specifically, instead of fitting the model to one time series of observed incidences among all patients, we fitted the model to two time series representing the observed incidences within and after 2 days of patients' admission. The area in the parameter

space with a high likelihood was reduced by rearranging the observation data, indicating an improved parameter identifiability. This is an example of what can be achieved by an informed segmentation of observations. Other possible segmentations (for example, by ward) and calibration of the likelihood can further improve identifiability. Another option is to calculate indicators of the spread of observations within the population. In our example, we might calculate the mixing entropy of the distribution of observations between wards (entropy is a common measure of the randomness of dispersion). This type of observation might improve identifiability if there is more nosocomial infection than importation, in which case we would expect observations to be more clustered. A more definitive examination of these data manipulation approaches is an expected part of future work.

The level to which a modeler needs to distinguish between different parameters is dependent on the questions being addressed. For example, in our system we are unable to identify the unique true parameters that give the correct fraction of nosocomial transmission; however, if we are simply interested in the qualitative question of the role of importation in MRSA incidence, the results for the synthetic outbreak in Fig 4C give a consistent answer that the majority of ($>$50%) cases are imported. Therefore, a useful definition of identifiability might be the ability to exclude enough of parameter space (i.e. reduce uncertainty in estimated parameters) such that the quantity in which we are interested is sufficiently constrained.

## Supporting information

**S1 File. Pseudo-code.** Pseudo-codes for algorithms used for computation in this paper. (DOCX)

## Author Contributions

**Conceptualization:** Tal T. Robin, Jeffrey Shaman, Sen Pei.

**Formal analysis:** Tal T. Robin.

**Funding acquisition:** Jeffrey Shaman, Sen Pei.

**Methodology:** Tal T. Robin, Jaime Cascante-Vega, Jeffrey Shaman, Sen Pei.

**Supervision:** Jeffrey Shaman, Sen Pei.

**Visualization:** Tal T. Robin.

**Writing – original draft:** Tal T. Robin, Jeffrey Shaman, Sen Pei.

**Writing – review & editing:** Tal T. Robin, Jaime Cascante-Vega, Jeffrey Shaman, Sen Pei.

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
