## [Decision Letter · Decision Letter 0]

7 Mar 2023

PONE-D-23-01761System Identifiability in a time-evolving agent-based modelPLOS ONE

Dear Dr. Robin,

Thank you for submitting your manuscript to PLOS ONE. After careful consideration, we feel that it has merit but does not fully meet PLOS ONE’s publication criteria as it currently stands. Therefore, we invite you to submit a revised version of the manuscript that addresses the points raised during the review process.

We look forward to receiving your revised manuscript.

Kind regards,

Martial L Ndeffo-Mbah, Ph.D

Academic Editor

PLOS ONE

Journal Requirements:

   "This study was supported by funding from the National Institutes of Health grant R01AI163023 (JS), Centers for Disease Control and Prevention U01CK000592 (JS, SP), Council of State and Territorial Epidemiologists NU38OT00297 (SP) and a gift from the Morris-Singer Foundation (JS)."

   "sponsors or funders did not play any role in the study design, data collection and

analysis, decision to publish, or preparation of the manuscript"

Additional Editor Comments:

Thank you for submitting your manuscript for consideration in PLoS ONE. During peer-review, a number of important issues have been identified stemming from clarifying the objective of the study to the robustness of the analysis to adequately address identifiability concerns. We would like to invite you to submit a revised version for your manuscript that thoroughly address the reviewers concerns as detailed in their reports.

Reviewers' comments:

Reviewer's Responses to Questions

**Comments to the Author**

1. Is the manuscript technically sound, and do the data support the conclusions?

Reviewer #1: Partly

Reviewer #2: Partly

2. Has the statistical analysis been performed appropriately and rigorously? 

Reviewer #1: Yes

Reviewer #2: N/A

3. Have the authors made all data underlying the findings in their manuscript fully available?

Reviewer #1: No

Reviewer #2: No

4. Is the manuscript presented in an intelligible fashion and written in standard English?

Reviewer #1: Yes

Reviewer #2: No

5. Review Comments to the Author

Reviewer #1: I have found the paper interesting. It is an excellent exhibit of the challenge of the lack of identifiability in a practical example. Besides that, the stochastic agent-based model application looks like a relevant contribution.

Mayor issues:

1- The abstract talks about an approach to discern whether model parameters have been properly identified and about a suggestion for data manipulation. After this, I continued reading, looking for some general description of the procedures and the ambit of application, which I could no find.

2- I feel that the term identifiability is used to refer to so many concepts that, at the end, it is dificult to keep up. For example, It is said that one of the issues that can affect the identifiability of a system is the identifiability of the model, wich I understand relates to structural identifiability, and when talking about the data the term practical identifiability, which has also been mentioned, is not used.

Minor issues:

3- (Line 126) "we will asume knowledge of two parameters, α and α" the same parameter is mentioned.

4- (Line 138) The reference to Figure 1 is wrong.

5- (Line 194) The term "demonstrated" might lead to the thought that a demonstration has been developed rather than an interpretation of results.

6- (Lines 300 and 301) The subtraction is divided into two lines which seems a bit confusing.

Reviewer #2: The manuscript addresses the issue with identifiability in agent-based models of infectious diseases that are calibrated to real-world data. The authors developed a simple agent-based model of Methicillin-resistant staphylococcus aureus (MRSA) using only four parameters. Two parameters were based on literature estimates (decolonization rate and daily probability of detection), while two parameters were calibrated (nosocomial transmission rate and importation probability). The authors acknowledge that this isn't a new concept and that researchers in the field are aware of the identifiability risks in the models, but they showed a case study for a specific disease. The authors concluded that although the model could match the data with their inference framework, multiple combinations of parameters can reproduce the same data. I found the point of the manuscript confusing. Below my main concerns.

1) I'm not sure what the goal of the study is. Is the goal of the study to "systematically evaluate" the issue of identifiability or to show a case study where this issue matters? From the title, abstract, and introduction, I got the sense that the point was to evaluate the issue. But, from the methods and results, I got the sense that this is a case study where the authors show the implications of unidentifiable parameters in an ABM calibrated to MRSA data from NYPH. From lines 215 to 225, the authors themselves outlined some solutions to make the parameters identifiable, but did not explore those options in their methods. Another potential solution to this issue would be to provide priors to the importation and transmission rate. How would that change the results?

2) I think the experiments are not necessarily enough to support the authors' conclusions. In my perception, it is not as important to determine whether an interplay of parameters might lead to parameters being unidentifiable as much as what are the requirements for those parameters to be identifiable. I suggest that the authors design additional experiments to determine these regions.

3) The authors did not provide enough motivation about the issue with those parameters being unidentifiable. For instance, what were the issues with those parameters being unidentifiable? Did the trajectories diverge in predictions? If the goal of a model is to project the disease dynamics, would that matter? As the authors mentioned, one issue could be the implementation of interventions targeted to a specific parameter. I encourage the authors to perform such simulations to determine the impact of unidentifiable parameters on public health interventions.

4) The structure of the manuscript could be better organized. Several sections in the results should be part of the methods.

** minor comments

- I think there is a lot of information about the algorithms that could probably go to the supplement.

- In the introduction, the authors claim that parameter identifiability is often disregarded in modeling studies, but they don't show examples of that

- In line 81 the authors mention that ABMs are increasingly used in real-world applications but don't mention the literature.

- Line 82. Is this study considered a systematic evaluation?

- Lines 122-124. The authors give a lot of unnecessary descriptions of stochastic models.

- Time tense: sometimes future sometimes past. A bit confusing.

- Line 140. What does "free simulation" mean?

- Lines 167-168. Why did the authors not perform scenarios of different targeted interventions?

6. PLOS authors have the option to publish the peer review history of their article (what does this mean?). If published, this will include your full peer review and any attached files.

Reviewer #1: No

Reviewer #2: No

---

## [Author Response · Author response to Decision Letter 0]

21 Apr 2023

Reply to reviewers is also added in a separate file. 

Reviewer 1: 

1. The abstract talks about an approach to discern whether model parameters have been properly identified and about a suggestion for data manipulation. After this, I continued reading, looking for some general description of the procedures and the ambit of application, which I could not find.

Reply: 

We thank the reviewer for this comment. In our results (Lines 174-181 and Figure 5 in previous version) we show that the inference algorithm is unreliable in retrieving the correct parameters. However, we agree this message was not brought forth strongly enough and therefore made the following changes to the revised manuscript. Please see revision in lines 244-253 in revised manuscript.

In the discussion (Lines 215-226) we provide suggestions to improve identifiability. We are currently working on implementation and examination of these suggested approaches. This work is involved, and it is our intention to publish these results separately. 

I feel that the term identifiability is used to refer to so many concepts that, at the end, it is difficult to keep up. For example, It is said that one of the issues that can affect the identifiability of a system is the identifiability of the model, which I understand relates to structural identifiability, and when talking about the data the term practical identifiability, which has also been mentioned, is not used.

Reply: The classification of identifiability into structural and practical categories, while established for deterministic models is less well defined for stochastic models where structural and practical can overlap; Instead, we identify 4 sources affecting identifiability in evaluating our system, as discussed on Lines 263-285. We feel this classification captures the processes at play for this system and have clarified this in the revised manuscript.

2. (Line 126) "we will assume knowledge of two parameters, α and α" the same parameter is mentioned.

Reply: This typo has been fixed, thanks for bringing it to our attention. 

3. (Line 138) The reference to Figure 1 is wrong.

Reply: Mislabeling fixed, thanks for bringing it to our attention. 

4. (Line 194) The term "demonstrated" might lead to the thought that a demonstration has been developed rather than an interpretation of results.

Reply: We changed the wording from “demonstrated” to “show”. Thank you for your remark. 

5. (Lines 300 and 301) The subtraction is divided into two lines which seems a bit confusing.

Reply: We agree that the alignment of the equation was confusing, as well was the phrasing of the entire paragraph. Please see lines 157-168 in the revised manuscript. 

Reviewer 2: 

1. I'm not sure what the goal of the study is. Is the goal of the study to "systematically evaluate" the issue of identifiability or to show a case study where this issue matters? From the title, abstract, and introduction, I got the sense that the point was to evaluate the issue. But, from the methods and results, I got the sense that this is a case study where the authors show the implications of unidentifiable parameters in an ABM calibrated to MRSA data from NYPH. From lines 215 to 225, the authors themselves outlined some solutions to make the parameters identifiable, but did not explore those options in their methods. Another potential solution to this issue would be to provide priors to the importation and transmission rate. How would that change the results?

Reply: In our paper we are aiming to illustrate a broader concept using a simple case study: here a stochastic ABM informed by time-varying real-world data. However, the issues that we are discussing are not unique to the example we provide. By using the agent-based model of MRSA colonization in NY hospitals, we demonstrate the broader issue of how stochasticity and model structure can conspire to undermine identifiability.

For this publication we limit the scope for two reasons. First, we are currently working on the implementation of these approaches and the scope of that discussion is larger than this paper and will be included in a future publication. Additionally, while we think that these suggestions are relatively general and are aimed at inspiring people to think of methods appropriate to their systems, the application to our model is too involved in the specifics of our system.

2. I think the experiments are not necessarily enough to support the authors' conclusions. In my perception, it is not as important to determine whether an interplay of parameters might lead to parameters being unidentifiable as much as what are the requirements for those parameters to be identifiable. I suggest that the authors design additional experiments to determine these regions.

Reply: Please see reply to the following question. 

3. The authors did not provide enough motivation about the issue with those parameters being unidentifiable. For instance, what were the issues with those parameters being unidentifiable? Did the trajectories diverge in predictions? If the goal of a model is to project the disease dynamics, would that matter? As the authors mentioned, one issue could be the implementation of interventions targeted to a specific parameter. I encourage the authors to perform such simulations to determine the impact of unidentifiable parameters on public health interventions.

Reply: Thank you for your remark. Figure 4 and the text supporting it were meant to stress the importance of being able to distinguish between different parameter sets in the equi-likelihood ridge. There, we show that though the parameters produce indistinguishable predictions for the observed variable (positive test number) and fail to predict hidden variables, such as total number of colonization and the ratio between nosocomial and community colonization cases However, we think that looking at the effect these discrepancies have on intervention would really bring home the message, and we thank you for suggesting this. We have added an example of an intervention (for example, intensified surface cleaning) represented as a proportional change in the nosocomial infection rate. Please see figure 5 in the revised manuscript for more details.

4. The structure of the manuscript could be better organized. Several sections in the results should be part of the methods.

Reply: We agree that the structure of the manuscript was not ideal. We have now moved the methods before the results and moved the algorithms to the supplementary. 

5. I think there is a lot of information about the algorithms that could probably go to the supplement.

Reply: Indeed, we agree. All algorithms were moved to the SI and some of the details about the calculations was removed all together. 

6. In the introduction, the authors claim that parameter identifiability is often disregarded in modeling studies, but they don't show examples of that. 

Reply: We find that it is a custom in our field and others that inference is conducted to only show convergence of the inference algorithm and to show that observations were reasonably reproduced (for example, many researchers use the “Pomp” package which produce parameter estimates but do not examine system identifiability). Here, we prefer to avoid pointing fingers to any specific publications and hope that people will take this message to heart regardless.

7. In line 81 the authors mention that ABMs are increasingly used in real-world applications but don't mention the literature.

Reply: Thank you for bringing this to our attention, we added some references. 

8. Line 82. Is this study considered a systematic evaluation?

Reply: We do not consider this work a systematic evaluation on its own; however, we do consider it the first step of a larger evaluation of identifiability issues specific to agent-based models and other intrinsically stochastic models. Our analysis contributes by offering a broader classification of the issues plaguing identifiability in such systems and by providing an example that is simple, addresses several issues, and is easily generalizable.

9. Lines 122-124. The authors give a lot of unnecessary descriptions of stochastic models.

Reply: While we agree that some readers might find this description trivial, we included it both because other readers might not and mostly to build the narrative of how identifiability is compromised by the nature of stochastic models.

10. Time tense: sometimes future sometimes past. A bit confusing.

Reply: We thank the reviewer for this editorial remark. We have transformed the relevant sentences to the past tense. 

11. Line 140. What does "free simulation" mean?

Reply: We used the term “free simulation” to describe a simulation in which parameters remain constant throughout the run time, to be distinguished from simulations for which parameters are adjusted due to the inference algorithm. In the revision, we decided to remove the word “free” and simply use ‘simulation’.

12. Lines 167-168. Why did the authors not perform scenarios of different targeted interventions?

Reply: While we initially avoided dealing with the issues of intervention due to the complexity in aptly describing its effects in theoretical models, the tried to comply with the reviewer suggestion and provide a schematic representation of an intervention (such that will reduce nosocomial transmission proportional to its native rate). This turned out to be a very elegant way to demonstrate the importance of the non-identifiability in this system. Please see figure 5 in the revised manuscript. We thank the reviewer for their suggestion.

---

## [Decision Letter · Decision Letter 1]

22 May 2023

PONE-D-23-01761R1System Identifiability in a time-evolving agent-based modelPLOS ONE

Dear Dr. Robin,

Thank you for submitting your manuscript to PLOS ONE. After careful consideration, we feel that it has merit but does not fully meet PLOS ONE’s publication criteria as it currently stands. Therefore, we invite you to submit a revised version of the manuscript that addresses the points raised during the review process. Please submit your revised manuscript by Jul 06 2023 11:59PM. If you will need more time than this to complete your revisions, please reply to this message or contact the journal office at plosone@plos.org. Please include the following items when submitting your revised manuscript:A rebuttal letter that responds to each point raised by the academic editor and reviewer(s). You should upload this letter as a separate file labeled 'Response to Reviewers'.A marked-up copy of your manuscript that highlights changes made to the original version. You should upload this as a separate file labeled 'Revised Manuscript with Track Changes'.An unmarked version of your revised paper without tracked changes. You should upload this as a separate file labeled 'Manuscript'.

We look forward to receiving your revised manuscript.

Kind regards,

Martial L Ndeffo-Mbah, Ph.D

Academic Editor

PLOS ONE

Reviewers' comments:

Reviewer's Responses to Questions

**Comments to the Author**

1. If the authors have adequately addressed your comments raised in a previous round of review and you feel that this manuscript is now acceptable for publication, you may indicate that here to bypass the “Comments to the Author” section, enter your conflict of interest statement in the “Confidential to Editor” section, and submit your "Accept" recommendation.

Reviewer #1: (No Response)

Reviewer #2: (No Response)

2. Is the manuscript technically sound, and do the data support the conclusions?

Reviewer #1: Partly

Reviewer #2: Yes

3. Has the statistical analysis been performed appropriately and rigorously? 

Reviewer #1: Yes

Reviewer #2: Yes

4. Have the authors made all data underlying the findings in their manuscript fully available?

Reviewer #1: Yes

Reviewer #2: Yes

5. Is the manuscript presented in an intelligible fashion and written in standard English?

Reviewer #1: No

Reviewer #2: Yes

6. Review Comments to the Author

Reviewer #1: I still think the paper has value for the scientific community, but I regret to inform you that the previous issues have not been satisfactorily addressed. Moreover, I have failed to properly carry out this revision due to the problems I will discuss below.

1. Saing ''We therefore propose a likelihood-based approach to discern whether model parameters have been properly identified'' still gives me the feeling that a general procedure will be described, while the paper is a particular application. Moreover, the cited lines in the response did not match the accessible material.

2. In this version a couple of the references appear to have errors and can not be properly seen.

3. The previous issue 4. (The term "demonstrated" might lead to the thought that a demonstration has been

developed rather than an interpretation of results) is not actualy adresed.

4. There exists at least a few differences between the version with the track of changes and the ones without.

The aforementioned problems make it so time-consuming to properly revise the paper that a could just make a partial revision which made me suggest a major revision.

Reviewer #2: I appreciate the effort made by the authors to address my comments. However, I think most of my comments were either avoided or ignored. Overall, the manuscript can benefit of clear references of previous studies to back up their claims, as well as improvement on their experiments' design and description.

In my first comment, I mentioned that the authors already outlined some references for methods developed to improve parameter identifiability, but there isn't a clear argument of the reason to not use those methods in this manuscript.

I also don't think comment 3 was properly addressed. The authors added a figure (Figure 5) in the manuscript but did not elaborate on the rationality for this experiment in the methods section. What were the authors expected to observe? Why this specific scenario?

Similarly, the authors mentioned that they " we prefer to avoid pointing fingers to any specific publications and hope that people will take this message to heart regardless." But I don't understand. How are the readers supposed to know that this is a real issue if it's not exemplified with references. I personally think that the way to illustrate an issue in a scientific manuscript is to provide references and not to expect readers to take the message to heart without evidence.

This issue comes up again in my comment #12. The authors added a whole new experiment, but did not describe this experiment in the methods.

7. PLOS authors have the option to publish the peer review history of their article (what does this mean?). If published, this will include your full peer review and any attached files.

Reviewer #1: No

Reviewer #2: No

---

## [Author Response · Author response to Decision Letter 1]

6 Jul 2023

Reviewer comments:

Reviewer 1: 

I still think the paper has value for the scientific community, but I regret to inform you that the previous issues have not been satisfactorily addressed. Moreover, I have failed to properly carry out this revision due to the problems I will discuss below.

Reply: We hope this new round of revisions satisfies your concerns.

1. Saying “We therefore propose a likelihood-based approach to discern whether model parameters have been properly identified” still gives me the feeling that a general procedure will be described, while the paper is a particular application. Moreover, the cited lines in the response did not match the accessible material.

Reply: While it is true that we described our approach in the narrow application of our model, it was done as an example of the utility of using synthetic trajectories for examining the identifiability of a system. We have rephrased the sentence to avoid misrepresentation:

“show that analyzing synthetic trajectories can support or contradict claims of identifiability. While we perform this on a specific model system, this approach can be generalized for a variety of stochastic representations of partially observable systems.”

 (See line 27 in the revised manuscript) 

2. In this version a couple of the references appear to have errors and can not be properly seen.

Reply: We thank the reviewer for bringing this to our attention. We have reentered all references and checked that the doi numbers refer to the correct papers. We hope this is now resolved. 

3. The previous issue 4. (The term "demonstrated" might lead to the thought that a demonstration has been developed rather than an interpretation of results) is not actually addressed.

Reply: In our initial response, we understood your remark as an objection to the extent to which a demonstration is performed and therefore tried to soften it. However, we now see that we misunderstood your meaning, and that your objection is to our claim that non-identifiability in this system is caused by an interplay between parameters, which is mere interpretation of the results. We agree that this is not a fact in the strictest sense, and therefore adjusted the text appropriately :

“Our findings can be reasonably interpreted as an example of this issue. The linear dependance between the inferred parameters along the ridge indicates that an interplay between the parameters may be the cause of unidentifiability.”

(See line 310 in the revised text) 

4. There exists at least a few differences between the version with the track of changes and the ones without. The aforementioned problems make it so time-consuming to properly revise the paper that a could just make a partial revision which made me suggest a major revision.

Reply: We apologize for the confusion. When we rearranged the location of the methods and some of the figures while track-changes was on, the Word file crashed. Therefore, we had to make these changes without tracking changes. The newly submitted file is tracked properly from the last submission and this should no longer be an issue.

Reviewer 2: 

I appreciate the effort made by the authors to address my comments. However, I think most of my comments were either avoided or ignored. Overall, the manuscript can benefit of clear references of previous studies to back up their claims, as well as improvement on their experiments' design and description.

Reply: Our apologies for this impression. Our intention was not to ignore or avoid your comments. We hope all comments are now properly addressed. 

1. In my first comment, I mentioned that the authors already outlined some references for methods developed to improve parameter identifiability, but there isn't a clear argument of the reason to not use those methods in this manuscript.

Reply: Our intention was to include a more robust assessment of these methods in a separate publication. However, to comply with the reviewer’s request, we include in the revised manuscript initial results from implementing observation segmentation. As can be seen in new Figure 7 and the corresponding text (lines 286-298 in the revised manuscript), using this method can produce some improvement in identifiability. However, it is important to state that this improvement is only exemplified, and the systematic advantage needs more extensive examination. 

2. I also don't think comment 3 was properly addressed. The authors added a figure (Figure 5) in the manuscript but did not elaborate on the rationality for this experiment in the methods section. What were the authors expected to observe? Why this specific scenario?

Reply: We thank the reviewer for insisting on these important points. Our rational for this experiment is to show that even though observed time series of incidence are indistinguishable under different parameters, outcomes of infection and colonization might differ once the system is perturbed by an intervention. We used a very simple representation to do so and chose a rather strong intervention to emphasize the point. We have adjusted the description of Figure 5 (lines 269-275 in the revised manuscript) in a way that we hope will clarify this issue. 

3. Similarly, the authors mentioned that they " we prefer to avoid pointing fingers to any specific publications and hope that people will take this message to heart regardless." But I don't understand. How are the readers supposed to know that this is a real issue if it's not exemplified with references. I personally think that the way to illustrate an issue in a scientific manuscript is to provide references and not to expect readers to take the message to heart without evidence.

Reply: While we are certain that the issue of too much trust being given to inferenced models without proper examination of identifiability is a widespread phenomenon and should be taken to consideration more often, we feel that it is not only unwise, but also unfair, to name specific people who fell to this practice which is so widespread. However, it is a fair point that when one is making a claim one must support it. Therefore, we are changing the claim that parameter identifiability is often not addressed to:

“the problem of non-identifiability in a stochastic system which, when overlooked, can significantly impede model prediction” in the abstract (line 19 in the revised text), and in the introduction to:

 “the issue of parameter identifiability, which, if not addressed properly can distort prediction and system certainty” (line 59 in the revised text) and provide two references to support this claim. 

4. This issue comes up again in my comment #12. The authors added a whole new experiment, but did not describe this experiment in the methods.

Reply: A section in the methods describing the experiment is now added (lines 175-196 in the revised text).

---

## [Decision Letter · Decision Letter 2]

17 Aug 2023

System Identifiability in a time-evolving agent-based model

PONE-D-23-01761R2

Dear Dr. Robin,

We’re pleased to inform you that your manuscript has been judged scientifically suitable for publication and will be formally accepted for publication once it meets all outstanding technical requirements.

Kind regards,

Martial L Ndeffo-Mbah, Ph.D

Academic Editor

PLOS ONE

Additional Editor Comments (optional):

Reviewers' comments:

Reviewer's Responses to Questions

**Comments to the Author**

1. If the authors have adequately addressed your comments raised in a previous round of review and you feel that this manuscript is now acceptable for publication, you may indicate that here to bypass the “Comments to the Author” section, enter your conflict of interest statement in the “Confidential to Editor” section, and submit your "Accept" recommendation.

Reviewer #2: All comments have been addressed

Reviewer #3: (No Response)

2. Is the manuscript technically sound, and do the data support the conclusions?

Reviewer #2: Yes

Reviewer #3: Yes

3. Has the statistical analysis been performed appropriately and rigorously? 

Reviewer #2: N/A

Reviewer #3: Yes

4. Have the authors made all data underlying the findings in their manuscript fully available?

Reviewer #2: Yes

Reviewer #3: Yes

5. Is the manuscript presented in an intelligible fashion and written in standard English?

Reviewer #2: Yes

Reviewer #3: Yes

6. Review Comments to the Author

Reviewer #2: I think my comments have been sufficiently addressed. Although, line numbering is still an issue with this review in some places.

Reviewer #3: This article is a description of non-identifiability of an agent-based model fit to a dataset of MRSA infection in set of hospitals. The question of identifiability analysis for stochastic and agent-based models is of interest and, to my knowledge, has not had best practices described for it. I am an infectious disease modeler with expertise in identifiability, though I primarily work in compartmental models. I know that POMP is often used for inference for stochastic models, though I don’t think it works for agent-based models, so examples of how agent-based modelers approach identifiability analysis are useful.

As a new reviewer brought in on this round, I see my job primarily as assessing whether the authors have been responsive to the previous reviewers’ comments and whether the article is scientifically sound as is. My answer is yes, I believe that the manuscript is of sufficient quality at this stage. Nevertheless, I will offer my comments/suggestions to the authors in case they are helpful.

1. I would have appreciated more explanation of why the inference procedure was appropriate and valid. Either this approach is established (in which case previous literature could be cited) or it is being proposed (in which case more explanation and justification should be offered). I was particularly confused by the updating of the parameter estimates over the trajectory (assimilation periods). In “Likelihood estimation” section, it seems to me that it is a trick to be able to estimate Pr(bar(O_j)|theta_i), but this trick is not really justified in the text or the ramifications explored. It seems to me that it is allowing the parameters to vary in time as long as they vary around a mean value – more discussion in the methods would be appreciated. I would personally have favored an approach directly calculating the likelihood over the whole trajectory of each simulation (averaging the likelihood over a large number of simulations) and then using MCMC or a sampling approach, but the authors are free to use any valid approach as long as they can justify it.

2. The first part of the results actually describes setting up of the simulation and should be in the Methods.

3. 300 seems a small number of simulations. I would have expected something more on the order of 10,000. Perhaps the simulations are particularly computationally intensive.

4. Figure 2. The colors described in the caption do not correspond to the colors that I see in the Figure.

5. The data represent an endemic trajectory, and endemic trajectories often have less information for parameters than the more dynamic epidemic trajectories. I think the authors should mention this in that section of the discussion.

6. There is not much text describing the specific results in the discussion. What does it mean that you can’t tell the difference between beta and gamma in the model? Show the reader why it makes intuitive sense. (Also, the choice of parameterization is a bit confusing – the canonical parameterization would have gamma and alpha switched).

7. PLOS authors have the option to publish the peer review history of their article (what does this mean?). If published, this will include your full peer review and any attached files.

Reviewer #2: No

Reviewer #3: No

---

## [Editor Report · Acceptance letter]

14 Sep 2023

PONE-D-23-01761R2 

System Identifiability in a time-evolving agent-based model 

Dear Dr. Robin:

I'm pleased to inform you that your manuscript has been deemed suitable for publication in PLOS ONE. Congratulations! Your manuscript is now with our production department. 

Kind regards, 

on behalf of

Dr. Martial L Ndeffo-Mbah 

Academic Editor

PLOS ONE